# The Relationship between Organizational Environment and Perpetrators’ Physical and Psychological State: A Three-Wave Longitudinal Study

**DOI:** 10.3390/ijerph19063699

**Published:** 2022-03-20

**Authors:** Gülüm Özer, Yannick Griep, Jordi Escartín

**Affiliations:** 1Department of Social Psychology and Quantitative Psychology, University of Barcelona, 08035 Barcelona, Spain; jordiescartin@ub.edu; 2Department of Work and Organizational Psychology, Behavioral Science Institute of Radboud University, 6500 HE Nijmegen, The Netherlands; yannick.griep@ru.nl

**Keywords:** workplace bullying perpetration, organizational trust, organizational justice, longitudinal, psychological distress, physical symptoms

## Abstract

Although job-related work environment studies found associations to workplace bullying perpetration, little work with longitudinal designs has been conducted on broader organizational measures, which may help design effective interventions for perpetration. Using a three-wave longitudinal design and drawing on Conservation of Resources Theory, we investigated whether organizational trust and justice predicted perpetration six months later. The sample consisted of 2447 employees from Spain and Turkey from various industries, such as services, manufacturing, and education. We also investigated whether physical and psychological health explained the relationship between organizational trust, justice, and perpetration. The results indicated that, in three months, organizational justice negatively predicted psychological and physical health deterioration, while unexpectedly, organizational trust positively predicted the same. Health conditions did not predict perpetration, in three months, while organizational conditions did not predict perpetration directly or indirectly in six months. Assessing and improving organizational trust and justice practices may help employee health improve over time. As organizational trust, justice, and health status are significantly related to current perpetration incidents, assessments of these subjects may be instrumental in identifying possible current perpetration phenomena.

## 1. Introduction

Workplace bullying is defined as a perpetrator’s systematic (e.g., weekly) and persistent (e.g., six months) negative behavior that harms others, mostly in subtle and discrete ways which are difficult to observe [1,2]. These negative behaviors can be “harassing, offending, socially excluding someone or negatively affecting someone’s work”, occurring “repeatedly and regularly (e.g., weekly) and over a period of time (e.g., about six months)”, and form “an escalating process in the course of which the person confronted may end up in an inferior position becoming the target of systematic negative social acts” [2] (p.18). 

Bullying assessments are mainly done using the self-labeling method, where employees are given a definition and asked if they experienced such a phenomenon, or by the behavioral method, where negative acts are listed for the employees to point out their occurrence. The negative acts that constitute bullying have been compiled under several scales, such as the Negative Acts Questionnaire (NAQ-R; [3]), and used widely in bullying literature to assess bullying [4]. Most of these scales assess the respondents’ perception of being exposed to negative behaviors by inquiring. These behaviors can include: control and manipulation of the work context of the target; control and manipulation of social activities and the physical workspace or the information given and received while carrying out the work tasks; abuse by offensive actions, such as attacking, injuring, and sneering at targets’ feelings and emotions; discrediting targets’ professional reputation and standing; belittling their knowledge, experience, efforts, performance; devaluation of the importance of the role of employees; or unjustifiably relieving them of their responsibilities or assigning them tasks that are useless, impossible, or inferior to their category in the organization.

If analyzed by method and geography, bullying prevalence in Mediterranean countries was 16.3–27.9% according to the self-labeling method, and was 10.1–16.0% according to the operative criterion of “at least once a week for at least six months.” Based on studies done in Eurasia using the self-labeling method, bullying prevalence was 2.7–13.0%, while based on the operative criterion method, it was between 4.6–22.0% [5]. Despite sustained high levels of this unethical practice in workplaces, popular TV shows continue to downplay workplace bullying and encourage joking practices that evolve into perpetration [6] (Sumner et al., 2016). Unfortunately, Human Resources professionals whom employees turn to for help tend to believe that bullying stems from interpersonal problems [7] and do not feel urged to act [8]. Upon formal bullying complaints, poor execution of investigations diminishes perceptions of organizational fairness and justice, triggering escalation to an outside party for further investigation [9].

Perhaps the least researched actors in workplace bullying situations are the perpetrators, revealing a lack of in-depth knowledge [2]. Few scales assess workplace bullying from the perpetrators’ perspective is [4], and the few studies that have measured perpetration prevalence [5] indicated that perpetration prevalence was around 9.5%. With high prevalence rates, workplace bullying is still a real organizational and societal problem, making the study of antecedents and mediators of perpetration imperative to inform and develop interventions.

Similarly, workplace bullying literature lacks comprehensive knowledge, specifically on perpetrators’ physical and mental health, how they perceive their organizational environment, and how they are affected by their acts. This gap in the literature encumbers effective interventions on active or potential perpetrators. Therefore, focusing on perpetrators, we examined the associations between work environment, behavior, and health states in order to explain perpetration. Our study will contribute to the workplace bullying literature in various ways. First, it broadened the scope by investigating perpetrators’ physical and mental health. Second, it combined work environment and personal factors. Third, the longitudinal design, with three data collection points 3 months apart, tested the temporal precedence of events. We also investigated reverse effects by testing how reports of perpetration impacted perpetrators’ health and their perception of organizational trust and justice over time. 

## 2. Theoretical Foundations and Hypothesis Development

Over the past 30 years, the Conservation of Resources (COR; [10]) Theory has become one of the most widely cited theories in organizational psychology. The basic principle is that individuals seek to obtain, retain, protect, and cherish those they value. These valuables are called resources and can be objects (e.g., car, tools for work), conditions (e.g., employment, tenure, seniority), personality characteristics (e.g., key skills and personal traits such as self-efficacy and optimism), and energy resources (e.g., credit, knowledge, money). Individuals in resource-rich environments are likely to accumulate resource gains while those in poor environments are likely to accumulate resource losses [11]. Individuals notice resource losses to be greater, quicker, and longer than resource gains. Additionally, the feeling of resource losses accelerates over time, possibly as an alarm mechanism for survival [12]. 

Theory suggests that psychological stress occurs when individuals’ resources are threatened with loss, with actual loss, or when individuals fail to gain sufficient resources following significant resource investment. Therefore, in cases of stress, individuals examine difficulties in overcoming stress and proactively adapt to environmental changes by preserving or renewing their resources for future use. COR has been used in an organizational context to predict a range of outcomes when faced with daily stressors draining resources. The theory was used to explain how leader-member exchange as a resource protected employees from engaging in counterproductive work behaviors [13], how loss of resources due to workplace bullying was related to presenteeism [14]; how loss of resources due to work stress was related to abusive supervision [15]; and how loss of resources due to work-family conflict distorted sleep [16]. 

In workplace bullying literature, the COR Theory was used to explain perpetration due to loss of resources while experiencing undermining and verbal abuse [17], task conflicts [18], a stressful work environment, inappropriate sense of humor, and being bullied [19]. 

Therefore, in line with COR Theory, we argued that the threat of or actual loss of resources (such as status, stable employment, acknowledgment and understanding from the employer, support from coworkers, etc.) created by organizations with poor organizational trust and justice, may trigger psychological stress and poor health conditions. When individuals exhaust their resources, they become defensive, strive to preserve themselves, and often act aggressively and irrationally. Therefore, they may strive to eliminate the threat by engaging in negative behaviors as a coping strategy toward others to protect themselves and thus gain resources. However, as such resource gains are perceived as smaller and slower than resource losses, the balancing act takes longer than the losses [12]. This leads to long-term, sustained negative acts, defined as workplace bullying.

### 2.1. Organizational Environment

The work environment hypothesis [20] suggests that work conditions (such as role conflicts, work overload, and job ambiguity) created by poor job design and an unfavorable social environment, foster bullying experiences. On the other hand, previous research also indicated that being a perpetrator of bullying may be initiated by ineffective coping, unsolved personal conflicts, and a poor organizational environment (The Three-way Model; [21]). Therefore, using the organizational aspect of the three-way model, we hypothesized that organizational trust and justice, mainly influenced by senior management, would precede the work context (i.e., job demands and resources) and would, in turn, predict employee health and perpetration events.

Previous studies showed that family business environments with balanced task–employee focus [22] were related to lower perpetration, while organizations with low psychosocial safety climates (PSC) were related to higher perpetration events [23]. On the individual and work unit level, reports of PSC were significantly and negatively linked to work unit emotional exhaustion via work–unit workplace bullying for perpetrators. PSC on work unit level was also an antecedent to work unit level reports of perpetration [24]. Researchers [25] analyzed the relationship between organizational change and perpetration and found no direct relationship between them, but the link was only established through psychological contract breach. The study showed that employees felt frustrated and betrayed after an organizational change if they perceived that the organization had not fulfilled its commitments (expected exchange of benefits) while they fulfilled theirs. The above results showed that workplace bullying seemed inevitable if organizations focused more on tasks and neglected employee well-being, health, and safety. Employees felt betrayed, not taken care of, or frustrated, and flared out against others in such conditions. If such behaviors are not condemned, they are learned and copied by many others, cultivating perpetrators who may also be victims of bullying themselves. Nevertheless, due to the scarce and contradicting results in work environment studies, it is still unclear how perpetrators are affected by the organization. 

One of business enterprises’ primary and common concerns is maintaining equity in labor relations where the economic and psychological balance between the employee and employer is fair. Employees seek an equitable balance between their contributions to the organization and what they receive in exchange [26]. Individuals who help out others and invest their time in the organization are more affected by experiencing bullying and thus have higher turnover intentions [27]. Performance-enhancing compensation practices designed to increase employee productivity may seem to be in sync with the Equity Theory. However, if perpetrators are triggered to achieve higher productivity, equity will be compromised, and productivity will eventually fall [28]. The Equity Theory also suggests that individuals who experience a situation that causes tension or distress will seek to reduce this tension and distress. Targets who receive persistent criticism start believing that they are ineffective in their work and submit to abusive leaders [29].

Based on Equity Theory [26], organizational justice is derived from perceptions of contributions made and outcomes obtained in exchange, as well as lines in the core of employee and employer relationships. While justice is an ongoing exchange of inputs and outputs, organizational trust builds up over time. Based on favorable conditions, employee trust continues as long as the organization meets the expectation of fairness. Due to its vital role in labor relationships, these two concepts are worth examination under workplace bullying. 

Researchers argued that the significant portion of dissatisfaction at work could be explained by the perception of injustice [30], and introduced the Equity Theory. Inequity was defined as the unequal ratios of outcomes to the input of the person and others, indicating the distributive justice exercised by the supervisors as perceived by the employees. Later on, procedural justice was introduced, referring to the fairness of the procedures in organizational outcomes [31], and interpersonal justice referred to the treatment people received [32]. All sorts of unjust perceptions increased individuals’ will to restore justice and act against mistreatment. Previous reviews indicated that organizational injustice was closely related to abusive behavior at work [33]. 

Organizational justice is the combination of employees’ perceptions of fairness in procedures, information sharing, and interaction among employees and how justice is distributed in the workplace [34,35]. Previous empirical studies showed that organizational justice reduced perceptions of workplace bullying [36] and improved employees’ physical health [37], while low organizational justice increased the risk of psychological distress [38]; justice instability added to physiological stress [39]; injustice was related to somatic health complaints [40], aggression [41] and revenge [42]. A previous conceptual study in workplace bullying literature argued that workplace injustice perceptions created a vicious cycle of bullying experiences, which led to poor perceptions of organizational justice for targets and bystanders [43]. Despite its suggested relationship to bullying, no published studies on organizational justice as an antecedent to perpetration were found. Organizational justice is closely and positively related to organizational trust [44,45]. Trust refers to the employees’ expectations from the organization or their belief that the organization will be beneficial to them (or at least not be harmful) in the future [46]. Previous research showed that poor perceptions of justice and trust negatively predicted aggression [47], but to our knowledge, no research has been conducted on organizational trust as another antecedent to perpetration. Therefore, we will examine how organizational justice and trust impact employee health and how these factors play a role in perpetration behavior. 

### 2.2. Physical and Psychological Health

Employee health can be measured by objective health markers or subjectively through physical and psychological symptoms. Many researchers established the relationship between employee health and being exposed to bullying, such as sleep troubles [48,49,50], mental health problems [51,52], and higher alcohol consumption [53]. Studies from abusive supervision literature showed that leaders’ depressive symptoms, anxiety, workplace alcohol consumption [54], and sleep deprivation [55] were related to their abusive behaviors, without exploring the causality. Previous longitudinal studies on perpetrators did not focus on physical and psychological health, except for one study [56] that found a marginal impact of personal vulnerability (i.e., depressive and anxiety disorder, psychological cases) on perpetration, 12 months. Therefore, not enough research has been done on perpetrators’ health and the causal relationships between their behavior and physical and psychological health. 

In terms of psychological health, one of the most common complaints is stress experienced at work. Stress activates individuals. If it is good stress (eustress), it involves feeling challenged, which might motivate individuals toward higher achievement. However, if it is bad stress (distress), it involves disturbing negative feelings and may result in avoidance or withdrawal. Psychological distress is an acute condition with a sudden onset and is the state of emotional suffering associated with situations that the individual has difficulty coping with in daily life and with negative feelings and thoughts [57]. Psychological distress can also be detected with a perceived inability to cope, change in emotional status, feelings of discomfort, and harming oneself, manifested as hopelessness, anxiety, depression, sadness, anger, hostility, fearfulness, neglect of appearance, and suicidal gestures [58]. Consistent with the stress theory, it has been shown by many research results that being bullied in the workplace causes psychological distress and subsequently leads to significant health problems and contributes to even more experiences of bullying [51,59]. Meanwhile, research showed that targets who had high levels of psychological detachment (recovery experiences) from daily work reported lower rates of bullying as a perpetrator [60]. 

Research on abusive supervision indicated a relationship between abusive supervision and distress. For instance: leaders’ psychological distress levels were related to subordinates’ distress, mediated by abusive supervision [61]; leaders’ distress predicted higher abusive supervision [54,62]; and abusive supervision experienced at work was related to spouse undermining at home, mediated by psychological distress [63]. While workplace bullying researchers have studied distress, only one rare research finding on accused bullies indicated their psychological distress [64] while their physical state was not studied. 

Therefore, we argued in this work that the unfavorable organizational environment may create stress and drain resources as employees experience distrust, injustice, continuous anxiety, and fear of possible job, position, organizational benefits, self-esteem, or power losses. The prolonged resource loss triggered due to poor organizational trust and justice may lead to psychological distress and physical symptoms related to distress. The eroding health status, coupled with poor organizational trust and justice, may create preconditions for the escalation of bullying. Therefore, we expect that (Figure 1):

**Hypothesis 1** **(H1).**Organizational Trust and Justice (Time 1) are negatively related to bullying Perpetration (Time 3).

**Hypothesis 2a** **(H2a).**Organizational Trust and Justice (T1) are negatively related to Psychological Distress (T2) and Physical Symptoms(T2).

**Hypothesis 2b** **(H2b).**Psychological Distress (T2) and Physical Symptoms (T2) are positively related to perpetration (T3).

**Hypothesis 3** **(H3).**The relationship between Organizational Trust and Justice (T1) and Perpetration (T3) is positively mediated by Physical Symptoms (T2) and Psychological Distress (T2).

## 3. Methods

### 3.1. Procedure

We collected data mainly by reaching out to two psychology and organizational psychology professors at Spanish and Turkish universities. We invited them to help the data-gathering phase by encouraging their students to find respondents who worked for at least 8 h per week, in line with the ILO definition of being an employed person [65], for the research in exchange for extra credit. Respondents were informed that the study was about employee health without explaining the hypotheses and disguising that it was bullying research. Students who brought ten respondents to the study could earn one extra course credit. Data obtained by students gathering respondents were heterogeneous and thus generalizable [66]. The study was also advertised on social media platforms (Facebook, LinkedIn, Instagram) in three languages.

Data were collected via the Qualtrics survey tool; ethical committee approval (IRB00003099) was obtained from the Universitat de Barcelona. All respondents provided electronic informed consent prior to completing the survey that they consented to participate in this study and to the use of their data for publication. They entered their email addresses to be contacted to complete the survey at Time 2 and Time 3. Although our theoretical framework did not provide information about the time frame, we used a minimum of six months to detect workplace bullying as per the operational definition. To decrease the attrition rate, we raffled ten gift vouchers for €25 and one Fitbit Inspire 2 fitness band for (70 €) among the respondents at each measurement moment. We finished the study in 11 months, keeping the intervals between data collection at a minimum of three months.

At Time 1, out of the 3663 responses, we conducted a data cleaning procedure and removed responses for: individuals not working for 8 h a day (*n* = 478), respondents not giving consent and leaving their email blank(*n* = 544), and respondents who completed the questionnaire multiple times based on email addresses (*n* = 52). As a result, we were left with 2589 respondents who completed the survey at Time 1. All the respondents who entered their email addresses were invited to complete the survey at Time 2 and Time 3. At Time 2, out of the 465 responses, individuals not working for 8 h a day (*n* = 68), respondents not giving consent (*n* = 10), and respondents who completed the questionnaire multiple times based on the email addresses (*n* = 11) were removed. As a result, we were left with 376 responses at Time 2 (response rate of 15% relative to Time 1). At Time 3, out of the 280 responses, individuals not working for 8 h a day (*n* = 42), respondents not giving consent (*n* = 6), and respondents who completed the questionnaire multiple times based on the email addresses (*n* = 13) were removed. As a result, we were left with 219 responses at Time 3 (response rate of 8% relative to Time 1). 

We deleted responses from respondents who changed their jobs between the different measurements (*n* = 52) as they may have influenced the lagged relationships [67]. We deleted responses (*n* = 90) from respondents who left their emails but did not participate in answering any other questions. We did not delete those left at Time 2 and 3, to avoid losing valuable information. Instead, we relied on the Full Information Maximum Likelihood (FIML) method to reduce the response bias [68]. When using FIML, missing values (either by not completing a full data collection wave or by just one item or a scale) are not changed or imposed, but missing data are processed within the analysis model. This method allowed the use of all available information to predict the model, and was superior to list-by-list deletion as no information was lost in the estimation of the analysis. The final sample included 2447 respondents.

### 3.2. Sample 

The sample consisted of 1319 women (54%) and 980 men (40%) with an average age of 34.5 (*SD* = 12.3); with an average company tenure of 7.3 years (*SD* = 8.9 years), working an average of 5.1 days of the week (*SD* = 0.86 days). The sample consisted of employees from different organizations within Spain (36.4%), Turkey (54.3%), and others (9.3%; mainly from the UK, USA, Belgium, Pakistan, Israel) across a wide range of economic sectors, such as services (15.1%), education (11.4%), health (9.3%), manufacturing (8.7%), and wholesale and retail trade (5.9%). 

Logistic regression analysis tested if participation in the three waves versus dropout after any point in time (coded as 1 for dropout; 0 for retention) was predicted by any demographics or study variables. Therefore, we conducted a logistic regression analysis on the 2447 respondents of T1, in which the outcome was participation vs. drop out, and the risk factors were: age, gender, supervisory position, and research variables (organizational trust and justice, psychological distress, physical symptoms, and perpetration scores). We found that younger respondents (OR: 0.96, *p* < 0.001) and those who were not supervisors (OR: 0.61, *p* = 0.01) were significantly more likely to leave the study. We also found that psychological distress was related to drop out, with those respondents experiencing less distress being significantly more likely to leave the study (OR:0.76, *p* = 0.007). To control for potential selection bias due to drop out, we examined whether respondents who dropped out (N = 2301) differed from the non-dropouts (N = 133) with respect to their demographic characteristics and levels on the study variables. As shown in Table 1, two samples differed regarding their age and supervisory position but did not differ significantly regarding the mean scores of study variables. This result of attrition from the study did not represent a threat to the external validity of our findings, since respondents’ study variable meant that those who dropped out were not significantly different from the respondents who participated for the remaining of the study. Therefore, there was no selection bias.

### 3.3. Measures

We adopted a complete panel design, in which we measured all variables at all three measurement periods [69]. All measures were administered in English, Spanish, or Turkish, and respondents were assured of confidentiality and informed that they could withdraw from the study at any point in time. To keep the attrition rate low, shortened scales were used as much as possible. To underline the period over which respondents were requested to report (i.e., three months), we reworded items such that they included the phrase “since the previous survey.”

Organizational trust (OT). The OT scale [46] consisted of seven items, an example being: “I believe my employer has high integrity.” Employees reported OT on a 7-point scale ranging from 1 (strongly disagree) to 7 (strongly agree). Reliabilities were satisfactory at all three measurement points: αT1 = 0. 85, αT2 = 0. 0.85, and αT3 = 0.87.

Organizational Justice (OJ). The OJ scale [34] consisted of seven items. An example item is: “Overall, I’m treated fairly by my organization”. Each item was assessed on a 7-point scale ranging from 1 (strongly disagree) to 7 (strongly agree). Reliabilities were satisfactory at all three measurement points: αT1 = 0.92, αT2 = 0.94, and αT3 = 0.94.

Psychological Distress (PD) scale [63] consisted of four items. At the beginning of the scale questions, the following introductory sentence was used: “In the past month, how often have you been feeling any of the following descriptions.” and ended with the statements such as “feeling fearful”. Answers were on a 7-point scale ranging from 1 (never) to 7 (always). Reliabilities were satisfactory at all three measurement points: αT1 = 0. 81, αT2 = 0.78, and αT3 = 0.81.

Physical Symptoms Inventory (PSI) was used to measure the health condition of employees as a mediator in the antecedent-outcome relationship. Being significantly related to the psychological state, the physical symptoms assessed physical and somatic health symptoms covering individuals’ digestive, visual, and central nervous system symptoms [70]. The full 13-item version of the PSI [71] (Duffy et al., 2019) was administered to ask respondents to rate how often they had experienced specific health symptoms over the past month. Items were rated on a 7-point scale ranging from 1 (never) to 7 (always). Reliabilities were satisfactory at all three measurement points: αT1 = 0. 86, αT2 = 0.85, and αT3 = 0.86.

Workplace bullying Perpetration. Perpetration was measured by the behavioral approach method by adopting the EAPA-T-R [72] to an active format. Respondents were asked to rate the following example behavior during the last six months; “I controlled or blocked correspondence, telephone calls or work assignments of others”, on a scale from 1 (never) to 7 (very frequently/more than once a week). As this construct shaped the behaviors of the respondents, it is unlikely that one would report engaging in perpetration with the same intensity over six months. Therefore, calculating the scale’s reliability became obsolete [25]. 

Age and tenure were measured in years. Gender was coded as 0 for female, 1 for male. The supervisory position was coded as 1 for being in a supervisory position and 0 for not being in a supervisory position. Sectors were coded according to *The Statistical Classification of Economic Activities in the European Community* (NACE codes). 

### 3.4. Analysis

Using a mediated SEM (structural equation modeling), we tested the relationship between organizational trust and justice perceptions at Time 1, psychological distress and physical symptoms at Time 2, and perpetration at Time 3. We estimated the indirect effect from perceptions of organizational trust and justice at Time 1 on perpetration at Time 3 via physical symptoms and psychological distress at Time 2 as the product of the relationship between the independent variable and the mediator and the relationship between the mediator and the dependent variable. 

As an initial step, we obtained skewness, kurtosis values, and histograms for our study variables (Table 2). We noticed that organizational trust (T1 skewness = −0.24, SE = 0.05; T2 skewness = −0.17, SE = 0.14; T3 skewness = −0.05, SE = 0.19) and organizational justice variables were (approximately) normally distributed (T1 skewness = −0.37 SE = 0.05; T2 skewness = −0.35, SE = 0.14; T3 skewness = −0.30, SE= 0.19). However, psychological distress was positively moderately skewed (T1 skewness = 1.24, SE = 0.05; T2 skewness = 0.91, SE = 0.14; T3 skewness = 1.14, SE = 0.19) as were physical symptoms (T1 skewness = 0.94, SE = 0.05; T2 skewness = 0.94, SE = 0.14; T3 skewness = 1.05, SE = 0.19). The workplace bullying perpetration variable was positively and highly skewed with unacceptable kurtosis levels at all times, which was expected (T1 skewness = 4.50, SE = 0.05; T2 skewness = 3.10, SE = 0.14; T3 skewness = 3.02, SE = 0.19). Therefore, we corrected perpetration for skewness by log 10 transformations of the variables [73] and used the log 10 transformed perpetration score in our analysis. We used this continuous measure of bullying (higher levels indicating higher bullying) for all the analyses. We also used the self-labeling measure of single item bullying question in COPSOQ III [74] and modified it to reflect enactment by defining bullying (e.g., bullying means that a person is repeatedly exposed to unpleasant or degrading treatment and that the person finds it difficult to defend himself or herself against it. Have you bullied others at your workplace in the last six months? 1 = Never, 7 = Very frequently). We correlated the modified EAPA-T-R scale with the modified single item bully score also transformed log ten. The correlation was r = 0.52, *p* < 0.01 at Time 1; r = 0.31, *p* < 0.01 at Time 2; r = 0.47, *p* < 0.01 at Time 3, thus supporting the construct validity of the measure. 

The data were collected mainly from Spain and Turkey with a few additions from countries in Europe, Asia, and the USA. We checked if the cultural differences impacted our outcome variable, which was perpetration, at T3. At T3, there were 105 responses from Turkey, 44 from Spain, and 5 from other countries. As “other country” data was negligible, the independent t-test was conducted between Spain and Turkey’s perpetration scores. There were no significant differences in the perpetration scores from Turkey (M = 0.05, SD = 0.10) and from Spain (M = 0.07, SD = 0.11)—conditions; t(147) = −1.29, p = 0.20. Therefore, cultural analysis was not conducted.

Data were analyzed using SPSS 26 (IBM Corp, New York, NY, United States) and SEM in AMOS 26.0 (Amos Development Corp, Wexford, Pennsylvania, United States) [75] (Arbuckle, 2019) based on maximum likelihood estimation. In evaluating the adequacy of models, we considered four fit indices: the chi-square, the comparative fit index (CFI), Tucker–Lewis index (TLI), and the root-mean-square error of approximation (RMSEA). When evaluating the goodness-of-fit of structural regression models with a chi-square value, a non-significant *p*-value indicates a good fit. However, in large samples, even small and substantively unimportant differences between the estimated model and the true underlying model will result in the test model’s rejection [76]. Consequently, other indices of model fit were also considered in this study. Based on stringent recommendations (Hu and Bentler, 1998), a CFI and TLI value of 0.90 or greater indicated a good fit, and values of 0.95 or greater represented excellent fits. The RMSEA point estimate indicated a good fit to the data at values of 0.10 or less, with values 0.06 representing excellent fits [77,78].

## 4. Results

Model fit was assessed using TLI, CFI, and RMSEA. A confirmatory factor analysis using Amos 26.0 was conducted to support the distinctness of the constructs of the variables measured in the study. Results of the confirmatory factor analysis suggested that the hypothesized five-factor model (organizational trust T1, organizational justice T1, psychological distress T2, physical symptoms T2 and perpetration T3) provided an acceptable fit to the data (χ2 = 2361 (485), TLI = 0.91, CFI 0.92, RMSEA = 0.04), better than all other possible models. Please refer to Figure 2a for a structural equation model of the mediation effects of physical symptoms and psychological distress on the relationship between organization and perpetration. 

### 4.1. Descriptive Statistics

The correlation (Pearson) analysis supported the relationships among the study variables. Table 3 shows the means, standard deviations, and inter-correlations between all variables under study at each of the three measurement points. The pattern of significant correlations was in the expected direction. Perpetration was lower in older respondents and higher among those in supervisory positions, but did not have any relationship with gender, contrary to previous findings on males more likely to be perpetrators [79,80]. 

### 4.2. Statistical Analysis 

In order to understand how organizations affected employees, significant correlations among the study variables were examined. As expected, initial organizational trust and justice (T1) were negatively associated with all three data points of psychological distress and physical symptoms. We also noted that initial organizational trust (T1) was negatively associated with perpetration at T1 and T2. However, the effect attenuated at T3, while initial organizational justice (T1) was associated negatively with initial perpetration; its effect on perpetration attenuated in later data collection points. Employee health in relation to perpetration was also examined. As expected, psychological distress and physical symptoms (T1) were positively associated with perpetration during the initial two data collection points, but effects attenuated at the third data collection. 

We tested the normal causation model, which included cross-lagged paths from organizational trust and justice at T1, the physical symptoms, and psychological distress at T2 to being a perpetrator at T3 (Figure 2b). The 5-factor model had a good fit (χ2 = 2467 (486), TLI = 0.90, CFI 0.91, RMSEA = 0.04). We also tested the reverse model to examine the cross-lagged paths from perpetration T1, the physical symptoms and psychological distress at T2, and perceptions of organizational trust and justice at work at T3. The reverse model did not fit the data (χ2 = 1241 (487), TLI = 0.85, CFI 0.87, RMSEA = 0.03), and we did not have data evidence to suggest reverse relationships.

The effects of the relationships in our model are summarized below in Table 4. Hypothesis 1 stated that organizational trust and justice (Time 1) are negatively related to perpetration (Time 3). Based on the results, Hypothesis 1 was rejected as organizational justice (*β* = 0.02, *p* = 0.1) and organizational trust (*β* = −0.01, *p* = 0.14) did not significantly predict perpetration (T3) directly.

Hypothesis 2a stated that organizational trust and justice (T1) are negatively related to psychological distress (T2) and physical symptoms (T2). As expected, organizational justice (T1) significantly and negatively predicted psychological distress (*β* = −0.59, *p* < 0.001) and physical symptoms (*β* = −0.63, *p* < 0.001). However, organizational trust significantly and positively predicted psychological distress (β = 0.34, *p* < 0.001) and physical symptoms (T2) (*β* = 0.35, *p* < 0.001, which was contrary to the expected negative direct effect. Therefore, Hypothesis 2a was partly confirmed. 

Hypothesis 2b stated that psychological distress (T2) and physical symptoms (T2) are positively related to perpetration (T3). Although they positively predicted perpetration, the effects were insignificant, and thus the hypothesis was rejected. 

Hypothesis 3 stated that the relationships between organizational trust and justice (T1) and perpetration (T3) are positively mediated by physical symptoms (T2) and psychological distress (T2). Results showed that organizational justice (T1) had an indirect and negative effect on perpetration (T3) (*β* = −0.67) as expected, while organizational trust (T1) had an indirect and positive effect on perpetration (T3) (*β* = 0.44), which was unexpected. As our data had missing values, the significance of the mediation results could not be tested. Therefore, to test the significance of the indirect relationships, missing data strategies were implemented in two steps.

Initially, all responses with any missing data were deleted. This listwise deletion eliminated an entire case of data, including data that were not missing. Therefore, after removing all responses with even one variable missing, a data set of 131 responses was formed. This subset of complete responses did not give a good fit to the data (*χ*2 = 872 (486), TLI = 0.83, CFI 0.85, RMSEA = 0.08). Therefore, we were unable to analyze the significance of the mentioned indirect effects, as the deletion resulted in statistical power loss with a large amount of data removed.

The second step was to use data imputation instead of removing all missing data. Regression imputation uses similar variables to calculate an estimate of the missing data and generally provides unbiased parameters [81], provided that the data are missing at random. Therefore, the dataset was screened for “missing data at random” using the Missing Value Analysis function in SPSS with Little’s MCAR test. Test results indicated that data were missing completely at random (*χ*2 = 30.259 (29), *p* = 0.40). Therefore, we used data imputation in AMOS to calculate the missing data by using regression imputation. In regression imputation, the model is first fitted using maximum likelihood, and linear regression predicts unobserved values [75]. Scores were imputed only for respondents who had complete data on at least 88% of the items in a total of 33 observable variables [82]. 143 individuals needed this procedure for the perpetration items at T3, and the imputed data set for 279 was formed. This subset of 279 responses gave a good fit to the data (*χ*2 = 764 (479), TLI = 0.94, CFI 0.94, RMSEA = 0.05). Therefore, we conducted the mediation analysis on this subset. Hypotheses 3 was tested by calculating bootstrapping confidence intervals using 2000 replications [83]. Our results revealed marginal effects; an indirect negative effect of organizational justice on perpetration (*β* = −0.0018, boot SE = −0.0025, 95% CI [−0.0064:−0.0002], *p* = 0.063) and the indirect positive effect of organizational trust on perpetration (*β* = 0.0005, boot SE = 0.0015, 95% CI [−0.0011:0.0035], *p* = 0.48). However, as both results were statistically insignificant, Hypothesis 3 was rejected, concluding that there was no mediation.

## 5. Discussion 

The purpose of this study was to expand the existing longitudinal literature on perpetrators, focusing on their health data and organizations. This study was the first to consider the causal relationship of organization trust and justice and perpetration. 

The positive relationship between justice and employees’ current physical health [40], long-term physical health [37], and current psychological health [38,39] were already established. As expected, the study showed a clear negative causal relationship over time between justice and employees’ physical and psychological health in three months. This result was consistent with the COR Theory, which explained the loss of resources (personal health) due to stress created in a poor organizational justice environment. 

Our study’s second and unexpected result was that organizational trust positively predicted psychological distress and physical symptoms over time. However, it is also important to note that our cross-sectional findings revealed a significant negative relationship between organizational trust and health data in all three waves, in line with previous cross-sectional studies on organizational trust and negative health perceptions [84], positive mental and physical health [85], and burnout [86]. Organizational trust might also have destructive consequences on employees. Employees who trust their organizations may work long hours, accumulating tangible (retirement benefits, stock options) and intangible (managerial positions) benefits. They may become too dependent on the organization if they perceive their employability as low or perceive that the cost of changing jobs may be too high due, given the accumulated benefits. In the motivated attributions model of trust development, researchers [87] suggested that there could be exaggerated evaluations of trustworthiness between two parties shaped by feelings of dependence. These could lead to irrational trust while failing to observe or actively discredit disconfirming evidence. For example, they mentioned the Stockholm syndrome, wherein individuals regard the other party as trustworthy to reduce the anxiety attached to their feelings of dependence. Therefore, one possible explanation could be that employees who worked for trusted organizations and managers worked harder and longer hours at the expense of their health where trust assessed could not overcome the negative effects of work conditions, especially during the pandemic (between May and December 2021). The Equity Theory [26,30] suggests that justice is a constant exchange of employees’ inputs and employers’ output, thus constituting an element of variability—whereas trust may be more stable, leading to higher injustice perceptions of employees in organizations they trust due to unfavorable working conditions during the pandemic. In terms of COR Theory, the trust they feel for the organization as a means of stable employment may add to their resources [10] (p. 342), balancing the resource drain caused by psychological distress due to the unjust environment. 

The third result we found was that, despite high negative cross-sectional correlations between health data and perpetration in times 1 and 2, psychological distress and physical symptoms did not predict perpetration over time (T2–T3). The COR Theory suggests that individuals defensively react to loss spirals by regrouping and waiting for help or offensively react by acting aggressively to change the conditions they are in as a coping mechanism. Therefore, when experiencing resource losses due to psychological distress created by the organizational environment, individuals may choose to wait for the situation to pass or act aggressively to change the situation for themselves. The role that time plays in resource losses and gains in the presence of acute versus chronic stressors is still examined by longitudinal studies. While there are effective ways of securing resource gains in short time frames, such as lunch breaks [88], some studies have shown that individuals adapt to stressors over time and do not lose resources over time [89]. Research suggests that individuals break from resource loss spirals by individual adaptation or social support [12]. Therefore, based on COR Theory, we may conclude that the individuals may have adapted to the organizational environment and thus did not show aggressive negative behavior in the long term, while shorter-term cross-sectional associations showed perpetration used as a coping mechanism.

Perpetration studies have long focused on the job and individual characteristics, while few studies examined broader constructs of organizations as antecedents. While no significant cross-sectional correlations were found between perceptions of organizational cultures, such as hierarchy, market, clan, and adhocracy, and reports of perpetration [90], a balanced people–task-oriented family firm environment was negatively correlated to perpetration [22]. We found significant and negative cross-sectional associations between organizational trust and justice and perpetration, confirming that perpetration was related to a poor work environment. However, we did not find any direct or indirect causal effects from organizational trust and justice on perpetration within a six-month time lag from T1 to T3. The link between organizational trust, justice, and perpetration was not established indirectly through employee health. Although causality was not established with the data, it seemed that low levels of organizational trust, justice, physical and psychological health coexisted with perpetration behavior. Therefore, future studies may utilize different scales to measure organization as a broader construct with different time lags to examine its association to perpetration. As perpetration is a complex phenomenon, another possible explanation is that other factors such as personality, team structure, or job characteristics may also influence the organization–perpetration relationship. 

Our findings could be used effectively to guide organizational interventions. The significant results of the present analyses suggested that assessments of organizational trust and justice would be a valuable strategy to identify departments and teams that could have adverse work conditions. Therefore, organizations may review and adjust their policies, practices, and procedures to provide fair and favorable work conditions, directly impacting employees’ health. Our results indicated that employees could be distressed and show physical symptoms even if organizational trust was established. Therefore, organizations should conduct further health assessments that could help detect vulnerable, overworked, stressed departments, teams, and individuals. 

A few limitations require mention, which may have impacted the results. First, the response rates of this study were low, with the response rate at T2 follow-up being 12% (303 over 2447) and (172 over 2447) T3 follow-up at 7%. Second, data were collected using self-report questionnaires, raising the possibility of common method variance and social desirability among respondents. Although anonymity was ensured, there was a possibility that individuals may have underreported perpetration. Such underreporting may have attenuated correlations between the variables. Third, this research collected data during pandemic conditions where lockdowns, remote working, or forced onsite working were in place. In particular, the work and health conditions of the employees might be experienced differently compared to a non-pandemic era. Fourth, we conducted the study with a minimum of 3 months’ time lag between waves and hence used the operational definition of bullying as negative acts occurring for at least six months or longer. Previous longitudinal research on organizational antecedents of perpetration (organizational change) have been consistent with regard to the time lag chosen [25]. Nevertheless, further studies with different time lengths could better capture the missed relations in this study. Finally, organizational justice and trust are highly correlated constructs. Although CFA indicated that trust and justice are separate factors, multicollinearity may have been the reason for organizational trust having a negative correlation but positive regression with health data. 

Future research would benefit from examining data from varying sources and across multiple periods. For example, combining self-reports with some form of objective data (e.g., coworker reports of perpetration, team-reported data for organizational trust and justice) may provide valuable insight to researchers. Objective measures may not capture the full range of employees’ perpetration behavior, but they can provide useful information, enable measurement triangulation, and provide additional evidence of the validity of self-report measures. Future research could continue to explore the association between the health status of perpetrators when examining why perpetration occurs in the workplace. Organizations may create perpetrators indirectly by destroying employee well-being. We tested the hypothesized model with two samples drawn from Spain and Turkey during the pandemic, where work conditions changed dramatically during the study span. Employees lost their jobs or stopped working due to lockdown implementation. We encourage researchers to replicate and extend our findings in samples drawn from different cultures and when work conditions are more stable in order to achieve higher retention rates. So far, the limited research published from the perpetrators’ perspective has applied a narrow range of moderators and mediators to explain the antecedent–perpetration relationship. Future research should study various antecedents, moderators, and mediators with perpetrators. As for research methods, research examining temporal precedence of events is rare; causality between many variables is still unknown. We measured changes in organizational trust and justice, physical and psychological health, and reports of perpetration in three data points. Going forward, the pattern of relationship between these variables at each point could also be assessed to examine possible changes in the pattern over time. Therefore, going forward, longitudinal studies on perpetrators with multiple data collection times, especially using diary methods and qualitative studies, should be encouraged. 

## 6. Conclusions

There were no statistically significant paths regarding organizational trust and justice to perpetration directly or indirectly through employee health. However, in line with COR theory, the threat or loss of resources (such as status, stable employment, acknowledgment and understanding from the employer, support from coworkers created by a poor organizational justice environment) seemed to cause psychological stress and poor health conditions over three months. No relation was found to suggest that individuals attempted to eliminate the threat by engaging in workplace bullying to protect themselves and gain resources. As COR theory also suggests, individuals may have adapted to the environment over time. Cross-sectional data demonstrated that the COR theory might be a fruitful approach to understanding the interdependences between perceived organizational factors (such as organizational justice), employees’ psychological and physical health, and perpetration. The present results shed light on possible prevention and intervention formulas that deserve further research attention. We hope this study will stimulate additional research into the role of workplace bullying perpetrators from their perspective. This will facilitate prevention and intervention mechanisms and programs to help them find more sustainable and ethical ways to cope with their work environment, generating positive and healthy workplaces for all employees without exclusion.

## Figures and Tables

**Figure 1 ijerph-19-03699-f001:**
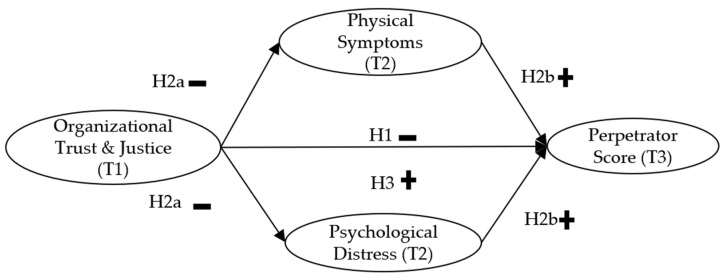
Overview of hypothesized relationships in the model.

**Figure 2 ijerph-19-03699-f002:**
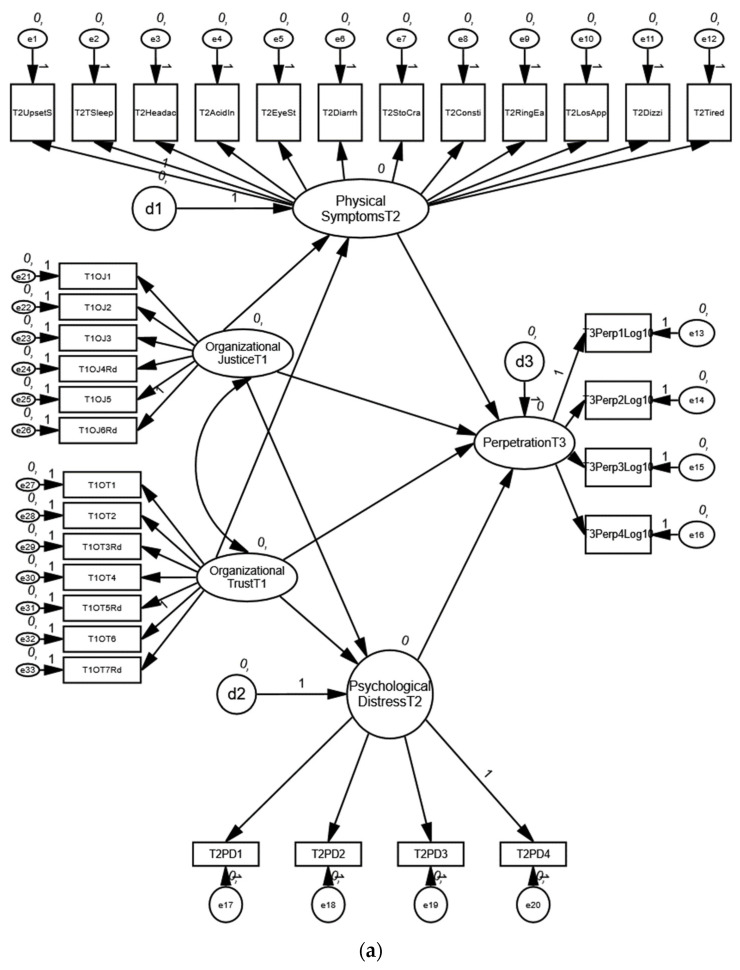
(**a**) Hypothetical structural equation model for mediation. (**b**) Structural equation model predicting perpetration (*n* = 2447).

**Table 1 ijerph-19-03699-t001:** T-test for Dropout Analysis.

Variables		N	Mean	*SD*	F	Sig.	t	df	P
Age	ND	126	40.02	10.93	9.23	0.00	5.22	2295.00	0.00
	D	2171	34.19	12.25					
Gender	ND	126	0.49	0.50	3.04	0.08	1.54	2297.00	0.12
	D	2173	0.42	0.49					
Supervisor	ND	126	0.43	0.50	28.58	0.00	3.96	2280.00	0.00
	D	2156	0.27	0.44					
T1OrgTrust	ND	133	4.69	1.35	0.81	0.37	−1.33	2432.00	0.18
	D	2301	4.84	1.27					
T1OrgJustice	ND	133	4.56	1.52	3.65	0.06	−0.89	2428.00	0.37
	D	2297	4.67	1.40					
T1PsyDistress	ND	132	2.56	1.17	1.84	0.17	1.80	2381.00	0.07
	D	2251	2.38	1.09					
T1PhySymptoms	ND	128	2.30	0.74	3.16	0.08	−0.25	2358.00	0.80
	D	2232	2.32	0.83					
T1Perpetration	ND	126	1.19	0.33	3.50	0.06	−0.68	2329.00	0.49
	D	2205	1.22	0.59					

Notes: *SD*: Standard Deviation, ND: Non-drop out, D: Drop out.

**Table 2 ijerph-19-03699-t002:** Skewness and Kurtosis.

Variables	N	Min	Max	Mean	*SD*	Skewness	*SD*	Kurtosis	*SD*
T1OrgT	2434	1.0	7.0	4.84	1.27	−0.24	0.05	−0.65	0.10
T2OrgT	303	1.6	7.0	4.66	1.22	−0.17	0.14	−0.57	0.28
T3OrgT	172	1.7	7.0	4.67	1.24	−0.05	0.19	−0.71	0.37
T1Ojus	2430	1.0	7.0	4.67	1.41	−0.37	0.05	−0.44	0.10
T2Ojus	298	1.0	7.0	4.60	1.37	−0.35	0.14	−0.41	0.28
T3Ojus	169	1.0	7.0	4.41	1.44	−0.30	0.19	−0.66	0.37
T1PsyD	2383	1.0	7.0	2.39	1.10	1.24	0.05	1.94	0.10
T2PsyD	293	1.0	6.0	2.41	1.01	0.91	0.14	0.86	0.28
T3PsyD	168	1.0	6.5	2.41	1.05	1.14	0.19	1.78	0.37
T1PhyS	2360	1.0	7.0	2.32	0.83	0.94	0.05	1.60	0.10
T2PhyS	293	1.0	4.9	2.24	0.76	0.94	0.14	0.61	0.28
T3PhyS	168	1.2	5.3	2.28	0.77	1.05	0.19	1.01	0.37
T1Perp	2331	1.0	6.0	1.22	0.58	4.56	0.05	25.60	0.10
T2Perp	290	1.0	3.5	1.17	0.38	3.10	0.14	11.14	0.29
T3Perp	168	1.0	3.0	1.16	0.36	3.02	0.19	9.87	0.37
T1PerpLog10	2331	0.0	0.8	0.06	0.13	2.72	0.05	8.40	0.10
T2PerpLog10	290	0.0	0.5	0.05	0.11	2.26	0.14	4.88	0.29
T3PerpLog10	168	0.0	0.5	0.05	0.10	2.32	0.19	5.17	0.37
Valid N (listwise)	125								

**Table 3 ijerph-19-03699-t003:** Descriptives and correlations.

	Variables	*n*	Mean	*SD*	1	2	3	4	5	6	7	8	9
1	Age	2297	34.51	12.26	-								
2	Gender	2299	0.43	0.49	0.01	-							
3	Supervisor	2282	0.28	0.45	0.21 **	0.17 **	-						
4	T1OrgTrust	2434	4.84	1.27	−0.05 *	0.05 *	0.05 *	-					
5	T2OrgTrust	303	4.66	1.22	0.00	0.03	0.08	0.69 **	-				
6	T3OrgTrust	172	4.67	1.24	0.06	−0.03	0.19 *	0.67 **	0.73 **	-			
7	T1Ojustice	2430	4.66	1.41	0.00	0.06 **	0.08 **	0.80 **	0.65 **	0.65 **	-		
8	T2Ojustice	298	4.60	1.37	0.03	0.06	0.18 **	0.68 **	0.82 **	0.69 **	0.71 **	-	
9	T3Ojustice	169	4.41	1.44	0.00	0.00	0.27 **	0.62 **	0.70 **	0.80 **	0.71 **	0.77 **	-
10	T1PsyDistress	2383	2.39	1.10	−0.15 **	−0.12 **	0.00	−0.41 **	−0.35 **	−0.37 **	−0.43 **	−0.35 **	−0.37 **
11	T2PsyDistress	293	2.41	1.01	−0.30 **	−0.16 **	−0.10	−0.23 **	−0.38 **	−0.38 **	−0.31 **	−0.40 **	−0.39 **
12	T3PsyDistress	168	2.41	1.05	−0.33 **	−0.10	−0.09	−0.35 **	−0.40 **	−0.40 **	−0.42 **	−0.44 **	−0.39 **
13	T1PhySymptoms	2360	2.32	0.83	−0.20 **	−0.21 **	−0.05 *	−0.27 **	−0.30 **	−0.38 **	−0.28 **	−0.28 **	−0.36 **
14	T2PhySymptoms	293	2.24	0.76	−0.29 **	−0.21 **	−0.12	-0.27 **	−0.34 **	−0.35 **	−0.31 **	−0.32 **	−0.34 **
15	T3PhySymptoms	168	2.28	0.77	−0.29 **	−0.20 *	−0.11	−0.34 **	−0.31 **	−0.39 **	−0.40 **	−0.33 **	−0.37 **
16	T1Perpetration	2331	0.06	0.13	−0.09 **	0.02	0.11 **	−0.10 **	−0.10	0.03	−0.09 **	−0.06	0.02
17	T2Perpetration	290	0.05	0.11	0.00	0.03	0.17 **	−0.15 *	−0.13 *	−0.16	−0.09	−0.16 **	−0.07
18	T3Perpetration	168	0.05	0.10	−0.18 *	0.06	0.06	0.08	0.04	0.01	0.11	−0.01	0.01
	Variables		Mean	*SD*	10	11	12	13	14	15	16	17	18
10	T1PsyDistress	2383	2.39	1.10	-								
11	T2PsyDistress	293	2.41	1.01	0.57 **	-							
12	T3PsyDistress	168	2.41	1.05	0.63 **	0.71 **	-						
13	T1PhySymptoms	2360	2.32	0.83	0.54 **	0.47 **	0.51 **	-					
14	T2PhySymptoms	293	2.24	0.76	0.50 **	0.60 **	0.59 **	0.73 **	-				
15	T3PhySymptoms	168	2.28	0.77	0.50 **	0.59 **	0.65 **	0.72 **	0.80 **	-			
16	T1Perpetration	2331	0.06	0.13	0.22 **	0.15 *	0.12	0.21 **	0.17 **	0.12	-		
17	T2Perpetration	290	0.05	0.11	0.15 *	0.14 *	0.19 *	0.22 **	0.21 **	0.24 **	0.33 **	-	
18	T3Perpetration	168	0.05	0.10	0.01	0.22 *	0.12	0.1	0.27 **	0.17 *	0.25 **	0.56 **	-

Notes: Gender: 0 = women, 1 = men; Supervisory position: 0 = not in supervisory position, 1 = in supervisory position; * *p* < 0.05; ** *p* < 0.01, Perpetration Score is Log 10 transformed.

**Table 4 ijerph-19-03699-t004:** Estimates, critical ratios, and standardized direct, indirect and total effects of the hypothesized model.

Structural Paths		Est	CR (*p*)	SRW	SDE	SIE	STE
OJ T1	PS T2	0.63	−5.13 (*p* < 0.001)	−0.85	−0.85	0.00	−0.85
	PD T2	−0.59	−5.09 (*p* < 0.001)	−0.93	−0.93	0.00	−0.93
	Perpetration T3	0.02	1.64 (*p* = 0.10)	1.04	1.04	−0.67	0.37
OT T1	PS T2	0.35	3.30 (*p* < 0.001)	0.55	0.55	0.00	0.55
	PD T2	0.34	3.46 (*p* < 0.001)	0.62	0.62	0.00	0.62
	Perpetration T3	−0.01	−1.50 (*p* = 0.14)	−0.72	−0.72	0.44	−0.28
PS T2	Perpetration T3	0.01	1.70 (*p* = 0.09)	0.44	0.44	0.00	0.44
PD T2	Perpetration T3	0.01	1.49 (*p* = 0.14)	0.32	0.32	0.00	0.32

Notes: OT: Organizational Trust; OJ: Organizational Justice; Physical Symptoms: PS; Psychological Distress: PD; Est = Regression weight estimates, CR = Critical ratio; SRW = Standardized regression weights, SDE = Standardized direct effects, IDE = Standardized indirect effects, STE = Standardized total effects.

## Data Availability

The data used in this manuscript can be found on the Open Science Framework using the following link: DOI 10.17605/OSF.IO/HR2ST.

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
