# Peer review of "The Relationship between Organizational Environment and Perpetrators’ Physical and Psychological State: A Three-Wave Longitudinal Study"

_ijerph, 2022, doi:10.3390/ijerph19063699_

Round 1
Reviewer 1 Report
I recommend the article for publication pending minor changes. The article is original, well-designed, based on an important new dataset and mostly well written. It seems to me that the main changes to be considered are the following:
(a) Clear designation that there was consent in collecting the data (even though I take it for granted that there was, given the size of the sample and the way the data were collected)
(b) A reformulation of certain passages to make them clearer. Here are some examples. First, CAT does not seem to do a lot of explanatory work in the paper. It is merely consistent with the (mostly expected) first result and seems to be inconsistent with the (unexpected) second result (i.e., the relationship between trust and negative impact on health). So its role should be reconsidered and the paper is significant in its findings anyway. Second, the main concepts used (bullying) should be better defined. Third, what was not clear to me (and perhaps requires some extra discussion) is how the questionnaire (question 7) managed to identify bullying perpetration (again, bullying should be better defined to make clear how that happens).
(c) There are some typos that have to be addressed (e.g. line 453 ‘[…] trust ad justice’ should be ‘trust and justice’).
Reviewer 2 Report
The authors took up a very important problem concerning relationships between some aspects of the work environment and a tendency to bully. Specifically, their focus is on organizational trust and justice perceived by employees, manifestations of physical and psychological health, and self-reported acts of perpetration. They use Cognitive Activation Theory (of Stress) as a theoretical framework to formulate study hypotheses. I must admit that the authors put a great effort into the realization of the project. They collected data in Span and Turkey; the total sample consisted of 2447 individuals; they reported their findings in a very detailed manner and attempted to explain results in a quite sophisticated way. However, there are significant flaws in the project on which I would like to comment:
- I am not sure whether CAT(S) is a proper theory for this project. Almost none of the key concepts of the CAT(S) are present in the study design, except Psychological Distress. The reasoning behind hypotheses is based mostly on common-sense speculations about consequences of conflict at work (which is not measured at all), feelings of threat (not measured as well), cognitive activation (not measured; what is that, first of all?), an expectation of conflict resolution (not measured). The most surprising conclusion is that an individual experiencing all of those events turns into a bully in the workplace. Why is that? It reminds a classic theory (Dollard & Miller, 1939; very much refined now, e.g., Berkowitz, Bandura) about links between frustration and aggression. But in any case, a theoretical framework has to be reflected in the choice of study variables, and hypotheses have to refer to the concepts of a given theory. So, my advice is to reformulate the theoretical background.
- The study is based on the data coming from two different countries (i.e., cultures) and unidentified “others”, and instruments were administered in three languages: English (this was for “others”?), Spanish and Turkish. First of all: why the country/culture variable has been ignored, and data from these countries lumped together? There must be differences between countries in organizational culture (see: Hofstede; collectivism – individualism; gender equality; power distance) that may affect the behaviors of employees, including bullying. So, cultural issues have to be elaborated on theoretically and included in statistical analyses.
- Also, in the case of the sample consisting of different national (cultural) subgroups, the first step before performing any statistical analysis has to be that of testing for measurement invariance. This testing tells whether the instruments employed in different (cultural, linguistic) groups can be interpreted in the same way and what statistical procedures are allowed to perform.
- The authors claim that the study is of a longitudinal design. As far as I understand, the idea of longitudinal studies is to examine the same individuals to detect any changes that might occur over a period of time. Indeed, the authors have measured each variable (justice, trust, physical symptoms, psychological symptoms, perpetration) at three points of time. One can expect that changes in these variables would be measured over time longitudinally, or – which will be more interesting - a pattern of relationships between variables at each point would be assessed regarding possible changes in a pattern. However, the authors constructed a model of relationships between variables in such a way that variables at one point were predictors of variables at other time points. I think it is an interesting approach but needs more explanation why it was chosen.
Reviewer 3 Report
Comments
- Introduction
Footnotes not APA-style, but square brackets with numbering.
The theoretical part has been properly worked out. It presents the most important information. I am only asking you to supplement with the most important data from the following articles:
https://doi.org/10.3390/soc11040143
https://doi.org/10.3390/ijerph17113851
https://doi.org/10.3390/ijerph16204039
Please expand on what is the study of reverse effects. (line 57)
- Theoretical Foundations and hypothesis development
As a complementary explanation, it is worth adding Conservation of Resources Theory
2.2. Physical and psychological health
The authors cite several studies, but it is also worth mentioning a few more recent ones
- Methods
Line 207 - some readers may be interested in what was the approval number of the ethics committee
Date cleaning was performed in a very good manner.
In my research, I also add a question of awareness: This is a question of vigilance. Please mark the number 2
Methods
3.3. Measures
This part is well described, nothing needs to be changed
- Results
What correlation coefficient did the authors use to measure gender and supervisory position with other variables?
- Discussion
The discussion was conducted in the correct way a reference to the most important cited publications. it is very important that the authors refer to Equity Theory (Adams, 1963). information about this theory should also be included in an introduction. Perhaps the author will find newer research on the topics they raise
Round 2
Reviewer 2 Report
I accept the revised paper. Thank you for your effort to make improvements.